# Robust and accurate prediction of residue–residue interactions across protein interfaces using evolutionary information

Sergey Ovchinnikov[1,2†], Hetunandan Kamisetty[1,3†], David Baker[1]*

[1]Department of Biochemistry, Howard Hughes Medical Institute, University of Washington, Seattle, United States; [2]Molecular and Cellular Biology Program, University of Washington, Seattle, United States; [3]Facebook Inc., Seattle, United States

**Abstract** Do the amino acid sequence identities of residues that make contact across protein interfaces covary during evolution? If so, such covariance could be used to predict contacts across interfaces and assemble models of biological complexes. We find that residue pairs identified using a pseudo-likelihood-based method to covary across protein–protein interfaces in the 50S ribosomal unit and 28 additional bacterial protein complexes with known structure are almost always in contact in the complex, provided that the number of aligned sequences is greater than the average length of the two proteins. We use this method to make subunit contact predictions for an additional 36 protein complexes with unknown structures, and present models based on these predictions for the tripartite ATP-independent periplasmic (TRAP) transporter, the tripartite efflux system, the pyruvate formate lyase-activating enzyme complex, and the methionine ABC transporter.

*For correspondence: dabaker@u.washington.edu

†These authors contributed equally to this work

Competing interests: The authors declare that no competing interests exist.

## Introduction

Recent work has demonstrated the accuracy of coevolution-based contact prediction for monomeric proteins using a global statistical model (*Thomas et al., 2008*) to distinguish between direct and indirect couplings (*Marks et al., 2011*; *Morcos et al., 2011*; *Hopf et al., 2012*; *Nugent and Jones, 2012*; *Jones et al., 2012*; *Lapedes et al., 2012*; *Marks et al., 2012*; *Sułkowska et al., 2012*; *Kamisetty et al., 2013*). While early approaches relied on estimating an inverse covariance matrix (*Marks et al., 2011*; *Morcos et al., 2011*; *Jones et al., 2012*), more recent studies have shown that a pseudo-likelihood-based approach (*Balakrishnan et al., 2011*) results in more accurate predictions (*Ekeberg et al., 2013*; *Kamisetty et al., 2013*) for a range of alignment sizes and protein lengths.

In contrast to this rich body of work for monomeric proteins, relatively little is known about the utility of such statistical models in predicting protein–protein interactions. The more general problem of predicting if two proteins interact with each other has been studied extensively using a wide variety of approaches (*de Juan et al., 2013*; *Hosur et al., 2012; Zhang et al., 2012*; *Shoemaker and Panchenko, 2007*, *Valencia and Pazos, 2002*, *Ochoa and Pazos, 2010*). Amino acid residue coevolution has been used to predict residue–residue interactions across interfaces with local statistical models (*Pazos et al., 1997*; *Halperin et al., 2006*). As noted above, the accuracy of these models is reduced by the confounding of direct and indirect correlations (*Lapedes et al., 1999*; *Weigt et al., 2009*); the application of global statistical models to coevolution-based contact prediction across interfaces has been limited to the case of the histidine-kinase/response-regulator two component system (*Burger and van Nimwegen, 2008*; *Weigt et al., 2009*; *Schug et al., 2009*; *Dago et al., 2012*).

**eLife digest** Proteins are considered the 'workhorse molecules' of life and they are involved in virtually everything that cells do. Proteins are strings of amino acids that have folded into a specific three-dimensional shape. Proteins must have the correct shape to function properly, as they often work by binding to other proteins or molecules—much like a key fitting into a lock. Working out the structure of a protein can, therefore, provide major insights into how the protein does its job.

Two or more proteins can bind together and form a complex to perform various tasks; and solving the structures of these complexes can be challenging, even if the structures of the protein subunits are known. Now, Ovchinnikov, Kamisetty, and Baker have developed a method for predicting which parts of the proteins make contact with each other in a two-protein complex.

Different species can have copies of the same proteins; but a copy from one species might have different amino acids at certain positions when compared to a related copy from another species. As such, when pairs of interacting proteins from different species are compared, there will be many positions in the two proteins that vary. However, if the amino acid at a position in one protein (let's call it 'X') varies, and the amino acid at, say, position 'Y' in the other protein also varies such that for any given amino acid at position Y there is often a specific amino acid at position X; positions X and Y are said to 'co-vary'. Ovchinnikov et al. noticed that when a pair of amino acids (one from each protein in a two-protein complex) co-varied, these two amino acids tended to make contact with each other at the protein–protein interface.

Ovchinnikov et al. used the new method to make predictions about the protein–protein interfaces in 28 protein complexes found in bacteria, and also to make a prediction about the interface between protein subunits in the bacterial ribosome. When these predictions were checked against the actual structures, which were all known beforehand, they were found to be accurate if the number of copies of each protein being compared is greater than the average length of the two proteins.

Ovchinnikov et al. went on to predict the amino acids on the protein–protein interfaces for another 36 bacterial protein complexes with unknown structures, and to present models for four larger complexes. The next challenge is to extend the method to protein complexes that are found only in eukaryotes (i.e., not in bacteria). Since the number of related copies for eukaryotic proteins tends to be smaller, there are fewer proteins to compare and it is therefore harder to detect 'covariation' when it occurs.

In this study, we examine residue–residue covariation across protein–protein interfaces using a pseudo-likelihood-based statistical method. In a large set of complexes of known structure, we find that covarying pairs of positions are almost always in contact in the three-dimensional structure, provided there are sufficient aligned sequences. We find further that significant residue–residue covariance occurs frequently between physically interacting protein pairs but very rarely between non-interacting pairs, and hence should be useful for predicting whether two proteins interact. We use the pseudo-likelihood method to predict contacts across protein-interfaces for 36 evolutionarily conserved complexes of unknown structure and present structure models for four of the complexes particularly well constrained by these data.

## Results

For a single protein family, it is straightforward to generate a multiple sequence alignment and subsequently identify covarying residue pairs. To identify covarying residue pairs between two proteins A and B is not as easy: only organisms that contain an ortholog of protein A and protein B contribute, and in generating the alignments the protein A and protein B sequences for each organism must be properly paired. To simplify the ortholog identification problem, we focus on pairs of genes with conserved chromosomal locations separated in the genome by fewer than 20 other annotated genes. We then build GREMLIN global statistical models for sequences in the paired protein families. The models have 'one-body' parameters for each amino acid at each position in the two proteins, and 'two-body' parameters for each pair of amino acids at each pair of positions in the two proteins. These parameters are obtained by maximizing the pseudo-likelihood of the observed sequence pairs,

rather than their likelihood, which makes the quite formidable estimation tractable. In the following sections, we investigate the structural contexts of residue pairs with large values of these two-body coupling parameters

## Residue–residue covariation in the bacterial 50S ribosomal unit

We began by studying residue–residue coupling parameters in the bacterial 50S ribosomal subunit— the largest evolutionarily conserved bacterial multiprotein complex with an atomic resolution structure. For each individual protein in the complex, we constructed multiple sequence alignments by querying the UniProt sequence database (*Wu et al., 2006*) for homologous sequences. For every pair of proteins in the complex, we then constructed a paired multiple sequence alignment ('Materials and methods'). For each such paired alignment, we built a GREMLIN global statistical model, computed normalized coupling strengths from the two body coupling parameters, and ranked inter protein residue pairs based on these scores ('Materials and methods'). A coupling strength larger than one indicates higher than average coupling between two residues.

We find that in the 50S ribosomal subunit only a small fraction of residue pairs coevolve, as indicated by coupling strengths (y axis of *Figure 1A*) greater than 1.5. Remarkably, the two residues in

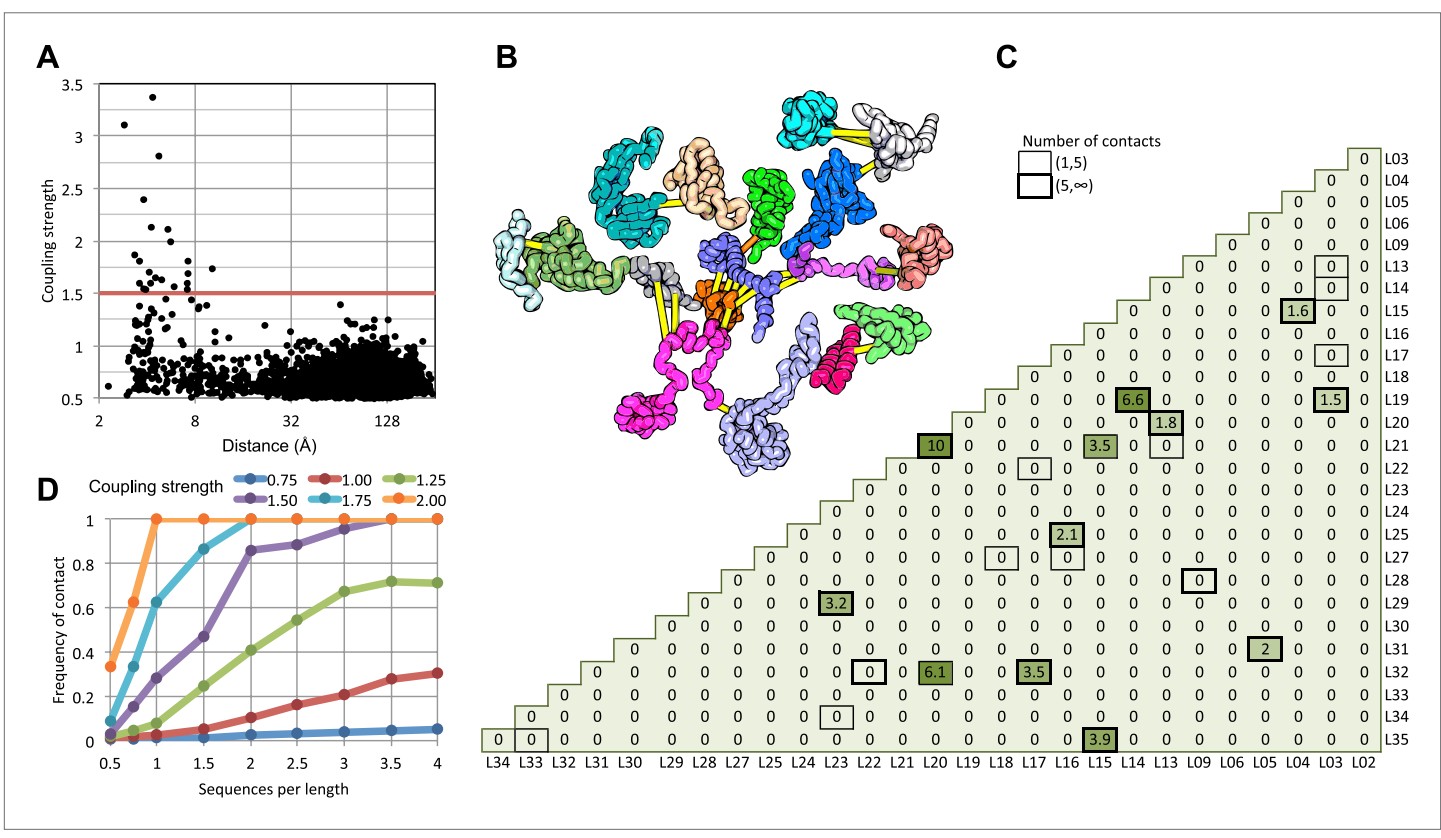

**Figure 1**. Residue pairs with high normalized coupling strengths are in contact in the 50S ribosomal subunit. (**A**) Coupling strengths and inter-residue distances for each residue pair in the 50S subunit (black dots). Residue pairs with coupling strength greater than 1.5 are nearly always less than 8 Å apart. (**B**) Locations of coevolving (high coupling strength) residue pairs in the protein component of the 50S subunit. The monomers have been pulled apart slightly for clarity. Lines connect residue pairs with coupling strength greater than 1.5; yellow, distance less than 8 Å; orange, distance less than 12 Å. (**C**) Protein pairs with strong inter-residue covariation (colors) make contact in the three-dimensional structure (black boxes). For each protein pair, the sum of the coupling strength greater than 1.5 for each pair of 50S subunit proteins is indicated; black boxes indicate contacts in the crystal structure. (**D**) Dependence of contact prediction accuracy on coupling strength and the number of sequences in the alignments. For each of the indicated coupling strength cutoffs (colors), the frequency of contact in the 50S structure (y axis) was computed for sub alignments with different sequence depths (x axis).

The following figure supplements are available for figure 1:

**Figure supplement 1**. Determining GREMLIN scores from normalized coupling strengths.

each of these pairs are almost all within 8 Å of each other in the 50S crystal structure (*Figure 1A*) and all are within 12 Å. The locations of the covarying residue pairs in the 50S structure (with the individual proteins pulled apart for clarity) are shown in *Figure 1B*; yellow lines indicate distances less than 8 Å and orange lines, distances less than 12 Å. For the 50S ribosome, the GREMLIN model was built using sequence data from ~1500 non-redundant genomes; *Figure 1D* suggests that for complexes with such large numbers of aligned sequence, residue–residue interactions across interfaces can be predicted with quite high confidence based on amino acid sequence covariation.

For a large protein–protein complex, can the sum of the coupling strengths between pairs of proteins in the complex be used to distinguish directly interacting and non-interacting protein pairs? In the 50S subunit, every pair of proteins with summed coupling strengths (numbers in *Figure 1C*) greater than 1.5 interacts with each other (boxes in *Figure 1C*). There are, however, several instances of protein pairs that contact in the 50S subunit for which no covariance is observed; clearly not every interaction will be identified by the sum of the coupling strengths, for example between two proteins that are held together primarily by the ribosomal RNA.

How many aligned sequences are required for accurate contact prediction? To assess the dependence on alignment depth, we generated paired sub-alignments with varying numbers of sequences for every pair of 50S proteins and recomputed coupling strengths for each sub-alignment. For each alignment depth, we calculated the fraction of residue pairs within 12 Å for different ranges of coupling strengths. We find that the greater the number of aligned sequences, the lower the value of the coupling strength above which residue pairs are likely to be in contact in the structure (*Figure 1D*). For example, if the number of aligned sequences is greater than the sum of the lengths of the two proteins, residue–residue contact predictions are likely to be accurate if the coupling strength is 2 or greater (*Figure 1D*: orange dots), while if there are twice as many sequences, contact predictions are accurate above a coupling strength of 1.5 (the cutoff shown in *Figure 1A*). A sigmoidal function of the coupling strength and the number of sequences per position in the complex accurately fits the observed contact frequency data ('Materials and methods' and *Figure 1—figure supplement 1*); we refer to the fitted values as GREMLIN scores for the remainder of the paper.

## Bacterial complex benchmark

We next generated paired-alignments for all *E. coli* gene-pairs that had conserved intergenic distances across genomes deposited in the UniProt ('Materials and methods'). As the 50S results (*Figure 1D*) suggested that alignment depths greater than the average of the lengths of the two proteins were required for accurate prediction, we focused on paired alignments with at least this number of sequences—1126 gene pairs in total excluding the ribosomal proteins. For each of these 1126 pairs, we generated GREMLIN global statistical models and determined the coupling strength for each residue pair.

For 64 of the 1126 gene pairs, at least one pair of residues had GREMLIN score >0.85. For 28 of the 64 pairs three-dimensional structures have been determined experimentally, and the locations of the residue pairs with GREMLIN score >0.6 for several of these complexes are shown in *Figure 2A* (pairs within 8 Å are in yellow, between 8 Å and 12 Å in orange, and greater than 12 Å, in red). Almost all pairs with GREMLIN scores greater than 0.6 are in contact in the complex structures, with the notable exception of the NADH dehydrogenase subunits (*Figure 2B*). The complex is thought to undergo a cascade of conformational changes during electron transfer (*Baradaran et al., 2013*); the high GREMLIN score contacts not made in the solved structure may provide insight into the nature of these changes. As observed for the 50S complex (*Figure 1C*), the existence of one or more high GREMLIN scores between two proteins provides evidence that the proteins interact: 44% (28/64) of the protein pairs with high GREMLIN scores form a complex which has been solved crystallographically compared to 8% (78/1126) over the whole set.

## Contact predictions for complexes of unknown structure

The results with the 50S ribosome and the protein pairs in the benchmark suggest that interactions can be accurately predicted across protein–protein interfaces given a sufficient number of aligned sequences. In *Figure 3*, we provide residue–residue contact predictions for the 36 of the 64 complexes with currently unknown structure (the *E. coli* gene sequences were clustered, and hence each complex may represent multiple *E. coli* gene pairs). These predictions should contribute to the determination of the structures of these biologically important complexes.

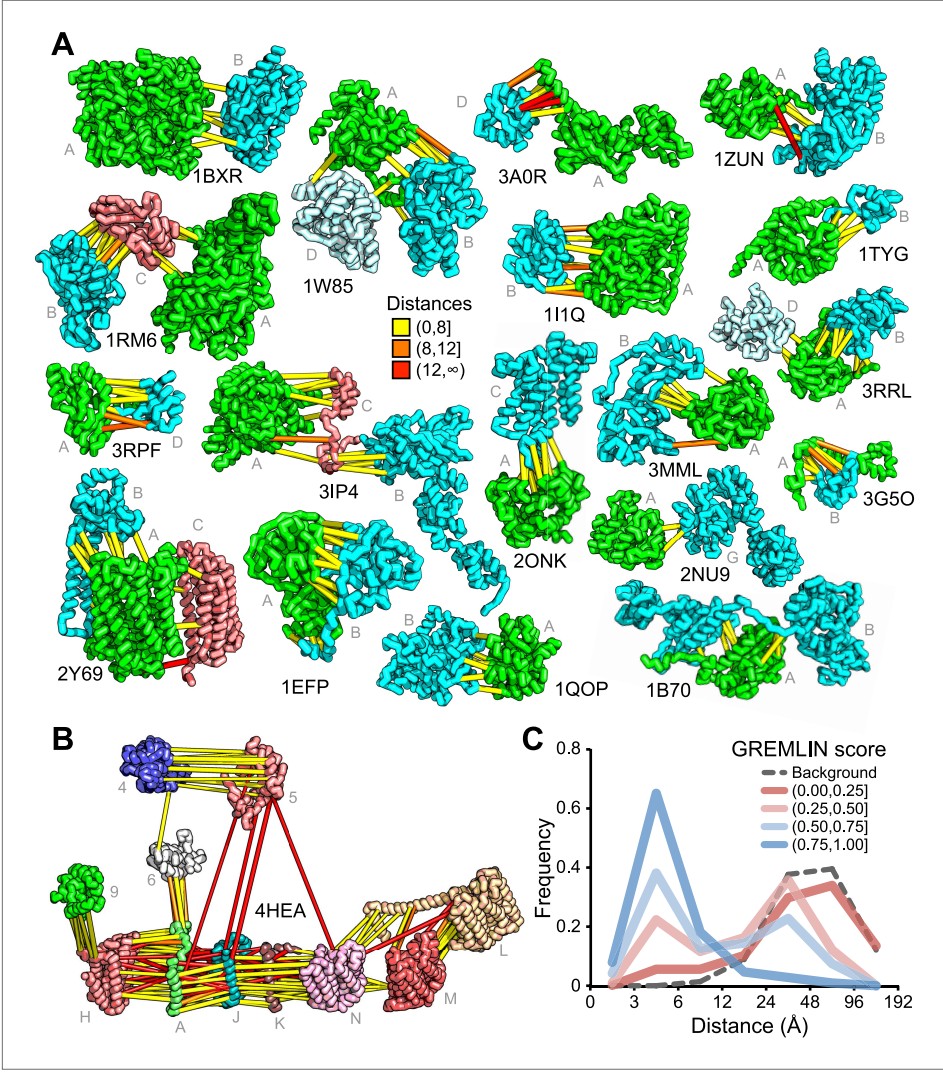

**Figure 2**. Residue covariation in complexes with known structures. (**A**) Residue-pairs across protein chains with high GREMLIN scores almost always make contact across protein interfaces in experimentally determined complex structures. All contacts with GREMLIN scores greater than 0.6 are shown; the structures are pulled apart for clarity. Labels are according to chains in the PDB structure. (**B**) Complex I of the electron transport chain has an unusually large number of highly co-varying inter residue pairs not in contact in the crystal structure of 4HEA; these contacts may be formed in different state of the complex. Residue pairs within 8 Å are in yellow, between 8 Å and 12 Å in orange, and greater than 12 Å, in red. Distances are the minimal distances between any side chain heavy atom. Labels are according to chains in 4HEA. (**C**) Dependence of inter-residue distance distributions on GREMLIN score. All residue–residue pairs between subunits in the benchmark set were grouped into four bins based on their GREMLIN score (colors), and the distribution of residue–residue distances (x axis) within each bin computed from the three-dimensional structures. See *Figure 2—source data 1* for the table of all the interfaces used in the calculation.

The following source data are available for figure 2:

**Source data 1**. PDB benchmark set.

## From contacts to structural models

Are the predicted contacts useful in assembling models of the protein complex from models of each component? We evaluated this on a docking test set containing 18 protein complexes from the benchmark set where at least one component (or a close homolog) had a known structure in the *apo* form ('Materials and methods', docking test-set). We developed a docking protocol that used the predicted contacts as distance restraints and sampled the space of physically plausible

structures to generate models of the protein–protein complex. The model with the best restraint score had an interface that was within 4 Å (in root mean square deviation) of the native interface in 14 of the 18 cases and had more than half the native contacts in 16 of the 18 cases (*Figure 4A*, *Figure 4—figure supplement 1*). Two of the cases in which the iRMSD (interface root-mean-square deviation) was the highest (bottom of table in A) are illustrated in *Figure 4B–C*: the high iRMSD is due to large changes in the conformation of one of the monomers upon binding; despite these changes the binding interface is reasonably accurately identified. Conformational changes that hinder the rigid-body docking protocol from sampling the bound conformation also occurred for thiazole synthase/sulfur carrier and phenylalanyl-tRNA synthase with iRMSD of 4.8A and 4.3A, respectively. In *Figure 4D*, a second energy minimum corresponds to a second interface in the complex with a different homo-oligomer subunit. In the absence of conformational changes, predicted contact guided docking is very accurate. The same protocol, on a positive control set of known bound structures of 41 protein-pairs (including 15 protein-pairs from the NADH electron transport complex), generated models that were within 2 Å of the native complex structure in 38 cases and within 4 Å in all but one case (*Figure 4—source data 1*, *Figure 4—figure supplement 2*).

Taken together, these results suggest that in cases with small conformational change, the docking protocol can recover the entire interface to high accuracy and in cases where binding is accompanied by a large conformational change, the protocol recovers the largest intact and/or unobstructed interface.

Of the complexes with unknown structure listed in *Figure 3*, we selected four cases with two or more high GREMLIN score (≥0.6) contact predictions across the interface that had experimentally determined structures for most of the subunits ('Materials and methods') and generated structural models of the complexes. These models provide the basis for formulating hypotheses about the structure/function of the complex, but we emphasize they are not experimentally determined structures; in particular the assumption in the modeling procedure that there are not large backbone rearrangements could be incorrect—in such cases the overall organization of the complex is still likely to be correct but the details of the interfaces could be considerably in error.

## The TRAP complex

The tripartite ATP-independent periplasmic (TRAP) transporters are composed of three proteins: two integral membrane proteins YIAM and YIAN, and one periplasmic protein YIAO (*Mulligan et al., 2011*). The structure of the periplasmic domain is known, but the membrane portion is unknown. To generate a model of the three-dimensional structure of the complex, we built YIAM models using Rosetta de novo structure prediction (*Simons et al., 1999*; *Raman et al., 2009*) guided by the intra-monomer predicted contacts, and models for YIAN and YIAO using RosettaCM comparative modeling. For YIAN the homologous structure of 4f35 (*Mancusso et al., 2012*) was used. The three monomer structure models were then assembled using PatchDock (*Duhovny et al., 2002*) and RosettaRelax (*Conway et al., 2014*) guided by the predicted intersubunit contacts ('Materials and methods'). In the resultant model of the complex (*Figure 5*), YIAO interacts with both of the membrane components; this is supported by a number of intersubunit contacts (yellow lines).

## Tripartite efflux system

Tripartite efflux complexes span both the inner and outer membrane, and are widely used in bacteria to pump toxic compounds out of the cell. The mode of interactions between the outer membrane factor and the membrane fusion protein is unresolved, with reports suggesting either a tip-to-tip interaction, the insertion of one into the other, or a multistage interaction with an initial tip-to-tip interaction, followed by sliding one through the channel of the other (*Long et al., 2012*). We generated homology models for the subunits based on the alignments to 1yc9 (*Federici et al., 2005*) and 3fpp (*Yum et al., 2009*) and docked them to generate models of the multidrug resistance protein complex. The predicted residue–residue contacts for this family of complexes support the tip-to-tip interaction (*Figure 4*; yellow lines); the coevolution data did not provide any evidence to support the insertion model.

## Pyruvate formate lyase-activating enzyme complex

Pyruvate formate-lyase (PFL) catalyzes the reaction of acetyl-CoA and formate from pyruvate and CoA in the Fermentation pathway. Formate acetyltransferase 1 or Pyruvate formate-lyase 1 (PFLB) is activated by Pyruvate formate-lyase 1-activating enzyme (PFLA). The structure of the complex is unknown, but the structures of the individual proteins have been solved (PDB ids: 3c8f [*Becker and Kabsch, 2002*] and 1h16 [*Vey et al., 2008*]). We carried out rigid body docking calculations with these two

**YIAM_YIAN** (3.9L)
| 21_F | 246_F | 1.00 |
|------|-------|------|
| 91_I | 25_L | 1.00 |
| 42_R | 52_D | 0.97 |
| 14_A | 268_L | 0.93 |
| 87_L | 26_L | 0.89 |
| 59_D | 273_K | 0.89 |
| 17_S | 245_A | 0.87 |
| 28_Y | 234_L | 0.84 |
| 43_Y | 48_V | 0.82 |
| 10_A | 268_L | 0.79 |
| 64_Q | 17_G | 0.76 |
| 56_A | 276_S | 0.75 |
| 9_L | 272_A | 0.75 |
| 46_V | 52_D | 0.72 |
| 13_L | 271_A | 0.67 |
| 50_F | 51_A | 0.66 |
| 95_L | 47_L | 0.62 |
| 46_V | 51_A | 0.60 |
| 10_A | 269_I | 0.60 |

**YIAN_YIAO** (3.6L)
| 168_S | 196_Y | 1.00 |
|-------|-------|------|
| 296_E | 62_K | 0.89 |
| 166_S | 221_E | 0.72 |
| 237_P | 197_T | 0.59 |

**YIAM_YIAO** (5.0L)
| 33_S | 186_N | 0.56 |
|------|-------|------|

**PFLA_PFLB** (0.7L)
| 66_F | 650_T | 0.92 |
|------|-------|------|
| 59_E | 643_K | 0.67 |
| 62_T | 570_N | 0.54 |
| 70_S | 688_G | 0.27 |

**MDTP_MDTN** (2.4L)
| 431_A | 146_E | 1.00 |
|-------|-------|------|
| 223_S | 146_E | 0.96 |
| 227_H | 142_P | 0.90 |
| 231_A | 150_S | 0.74 |
| 431_A | 148_F | 0.45 |
| 435_R | 142_P | 0.44 |

**METI_METQ** (2.0L)
| 80_R | 193_D | 0.99 |
|------|-------|------|
| 80_R | 192_D | 0.91 |
| 87_I | 182_L | 0.70 |
| 87_I | 66_F | 0.56 |
| 174_Q | 67_N | 0.50 |
| 87_I | 75_A | 0.43 |

**UMUC_UMUD** (1.0L)
| 415_S | 38_I | 1.00 |
|-------|------|------|
| 421_V | 54_V | 0.86 |
| 132_H | 76_S | 0.85 |
| 404_R | 33_Y | 0.72 |
| 183_D | 127_V | 0.66 |
| 78_A | 34_V | 0.60 |

**IF1_SECY** (1.4L)
| 32_V | 343_A | 0.89 |
|------|-------|------|

**YOHJ_YOHK** (1.9L)
| 64_C | 75_Y | 1.00 |
|------|------|------|
| 61_N | 79_E | 1.00 |
| 115_S | 94_I | 1.00 |
| 79_G | 38_M | 1.00 |
| 110_V | 101_V | 0.99 |
| 106_V | 101_V | 0.99 |
| 99_S | 115_A | 0.97 |
| 83_M | 42_I | 0.96 |
| 92_Q | 116_S | 0.93 |
| 104_T | 184_A | 0.89 |
| 105_L | 161_F | 0.84 |
| 40_I | 146_I | 0.84 |
| 71_M | 34_L | 0.83 |
| 65_Y | 79_E | 0.81 |
| 113_W | 169_M | 0.80 |
| 16_V | 14_I | 0.78 |
| 74_L | 65_L | 0.78 |
| 82_V | 42_I | 0.76 |
| 101_A | 157_L | 0.75 |
| 39_S | 70_V | 0.72 |
| 115_S | 176_A | 0.70 |
| 106_V | 105_T | 0.68 |
| 115_S | 179_L | 0.66 |
| 100_C | 150_C | 0.65 |
| 18_I | 72_A | 0.64 |
| 109_L | 169_M | 0.63 |
| 110_V | 97_I | 0.60 |
| 110_V | 213_I | 0.60 |

**CYOC_NUOK** (3.8L)
| 49_V | 80_L | 0.96 |
|------|------|------|
| 83_Y | 29_L | 0.68 |
| 75_L | 73_A | 0.68 |
| 65_E | 89_N | 0.65 |
| 101_W | 29_L | 0.61 |
| 42_I | 62_Y | 0.60 |

**ATP6_ATPF** (2.3L)
| 74_K | 34_E | 1.00 |
|------|------|------|
| 149_V | 10_Q | 0.93 |
| 77_T | 33_I | 0.92 |
| 53_G | 13_A | 0.86 |
| 155_L | 11_A | 0.78 |
| 50_V | 13_A | 0.72 |
| 255_I | 139_S | 0.68 |
| 243_I | 20_F | 0.64 |
| 49_S | 10_Q | 0.64 |
| 263_Y | 21_C | 0.64 |
| 239_V | 16_L | 0.63 |
| 111_W | 10_Q | 0.62 |

**FECI_FECR** (3.3L)
| 162_E | 14_R | 0.96 |
|-------|------|------|
| 133_Q | 57_R | 0.95 |
| 158_A | 18_H | 0.61 |

**FTSI_FTSW** (2.0L)
| 39_L | 313_V | 0.98 |
|------|-------|------|
| 42_V | 312_V | 0.94 |
| 47_V | 309_Y | 0.80 |
| 43_A | 309_Y | 0.74 |
| 34_A | 357_L | 0.62 |

**FLGB_FLGC** (2.4L)
| 34_D | 13_A | 1.00 |
|------|------|------|
| 34_D | 107_S | 0.99 |
| 110_A | 117_E | 0.98 |
| 98_N | 11_G | 0.98 |
| 121_S | 129_T | 0.96 |
| 34_D | 103_V | 0.96 |
| 113_S | 108_A | 0.89 |
| 27_A | 114_A | 0.88 |
| 117_Q | 128_K | 0.80 |
| 111_D | 122_V | 0.78 |
| 113_S | 125_M | 0.71 |
| 27_A | 27_N | 0.63 |
| 131_M | 69_E | 0.61 |
| 23_Q | 37_P | 0.60 |
| 93_P | 45_K | 0.60 |

**FLIP_FLIQ** (2.2L)
| 185_I | 84_L | 0.99 |
|-------|------|------|
| 54_I | 55_I | 0.90 |
| 229_V | 77_V | 0.89 |
| 203_V | 24_L | 0.68 |
| 205_M | 66_G | 0.67 |
| 213_P | 55_I | 0.66 |
| 243_F | 72_L | 0.65 |
| 149_L | 67_P | 0.64 |
| 188_T | 80_L | 0.63 |

**FLGH_FLGI** (0.6L)
| 52_F | 133_V | 0.99 |
|------|-------|------|
| 82_L | 257_S | 0.59 |

**FLHB_FLIR** (1.1L)
| 181_A | 188_L | 0.98 |
|-------|-------|------|
| 185_A | 192_T | 0.64 |

**FLIP_FLIR** (1.5L)
| 78_L | 187_A | 0.90 |
|------|-------|------|
| 97_T | 13_L | 0.71 |

**MLAD_MLAE** (1.4L)
| 14_L | 13_G | 0.93 |
|------|------|------|
| 14_L | 24_G | 0.76 |
| 8_I | 6_L | 0.65 |
| 58_V | 6_L | 0.65 |
| 65_V | 219_F | 0.63 |

**CCMC_CCME** (0.8L)
| 49_Q | 104_R | 0.95 |
|------|-------|------|

**CCMA_CCMB** (1.3L)
| 95_E | 16_R | 0.87 |
|------|------|------|

**MREB_NIFU** (0.6L)
| 21_N | 57_R | 0.97 |
|------|------|------|
| 149_I | 108_I | 0.72 |

**QMCA_YBBJ** (1.5L)
| 145_E | 115_R | 0.91 |
|-------|-------|------|
| 36_R | 113_H | 0.72 |

**PTPC1_PTPD** (1.2L)
| 130_K | 58_K | 1.00 |
|-------|------|------|
| 210_L | 189_I | 1.00 |
| 34_I | 122_C | 0.98 |
| 126_D | 28_R | 0.98 |
| 232_G | 180_L | 0.93 |
| 123_T | 27_E | 0.86 |
| 180_H | 258_S | 0.85 |
| 125_A | 32_G | 0.85 |
| 134_T | 53_L | 0.72 |
| 125_A | 61_L | 0.66 |
| 104_S | 71_V | 0.64 |
| 12_L | 223_I | 0.60 |

**YADG_YADH** (5.4L)
| 97_Q | 22_R | 0.94 |
|------|------|------|
| 49_G | 93_E | 0.93 |
| 106_Y | 99_P | 0.84 |
| 90_N | 15_K | 0.70 |
| 97_Q | 18_H | 0.68 |
| 82_L | 97_V | 0.67 |
| 104_G | 14_A | 0.64 |

**RBSA_RBSC** (5.6L)
| 365_S | 188_Y | 1.00 |
|-------|-------|------|
| 112_K | 188_Y | 0.97 |
| 301_A | 211_L | 0.95 |
| 93_Q | 204_G | 0.79 |
| 371_E | 197_R | 0.66 |
| 105_E | 198_Y | 0.64 |

**APAG_PPIC** (0.8L)
| 68_V | 73_E | 0.97 |
|------|------|------|

**DDPB_DDPC** (8.7L)
| 148_W | 268_G | 1.00 |
|-------|-------|------|
| 279_L | 231_T | 1.00 |
| 20_G | 135_I | 0.99 |
| 327_Y | 180_L | 0.97 |
| 327_Y | 177_G | 0.93 |
| 233_E | 292_K | 0.91 |
| 221_R | 283_D | 0.91 |
| 229_E | 212_P | 0.83 |
| 28_I | 147_A | 0.77 |
| 222_Q | 217_Q | 0.75 |
| 13_G | 127_A | 0.73 |
| 268_L | 137_L | 0.67 |
| 126_R | 293_A | 0.65 |
| 143_S | 279_N | 0.65 |
| 140_T | 272_L | 0.63 |
| 24_I | 135_I | 0.62 |
| 233_E | 185_Y | 0.59 |

**ENGB_NDK** (1.0L)
| 193_L | 60_F | 0.87 |
|-------|------|------|

**CLPP_CLPX** (1.3L)
| 96_F | 271_R | 0.95 |
|------|-------|------|
| 40_E | 267_G | 0.72 |

**APPB_APPC** (1.2L)
| 91_I | 474_S | 1.00 |
|------|-------|------|
| 101_P | 68_A | 0.99 |
| 95_C | 481_L | 0.95 |
| 88_V | 473_F | 0.94 |
| 95_C | 477_M | 0.92 |
| 342_I | 98_M | 0.90 |
| 91_I | 477_M | 0.77 |
| 332_S | 90_D | 0.70 |

**RNFE_RNFG** (1.8L)
| 188_L | 160_V | 0.90 |
|-------|-------|------|
| 85_M | 27_N | 0.90 |
| 67_I | 12_L | 0.89 |
| 146_G | 171_T | 0.66 |

**NRFC_NRFD** (1.2L)
| 130_Y | 92_S | 0.93 |
|-------|------|------|
| 129_Q | 76_T | 0.79 |
| 124_L | 87_H | 0.69 |
| 154_F | 55_T | 0.63 |

**TOLQ_TOLR** (4.9L)
| 149_I | 27_V | 0.91 |
|-------|------|------|
| 177_A | 27_V | 0.91 |
| 177_A | 25_L | 0.90 |
| 177_A | 26_L | 0.84 |
| 139_Y | 19_V | 0.83 |
| 142_L | 20_P | 0.77 |
| 179_A | 24_V | 0.76 |
| 167_V | 27_V | 0.76 |
| 179_A | 26_L | 0.73 |
| 177_A | 18_I | 0.69 |
| 133_V | 102_N | 0.67 |
| 177_A | 19_V | 0.67 |
| 156_L | 19_V | 0.66 |
| 142_L | 22_L | 0.66 |
| 200_N | 11_D | 0.63 |
| 134_G | 19_V | 0.59 |

**ATKA_ATKC** (0.7L)
| 190_I | 30_G | 1.00 |
|-------|------|------|
| 448_P | 91_N | 1.00 |
| 249_A | 175_V | 0.99 |
| 186_A | 26_T | 0.99 |
| 183_L | 26_T | 0.84 |
| 249_A | 167_L | 0.79 |
| 11_T | 170_Y | 0.78 |
| 198_F | 24_L | 0.63 |
| 238_A | 83_G | 0.62 |
| 244_F | 55_I | 0.60 |

**FEOA_FEOB** (0.9L)
| 41_R | 378_F | 0.94 |
|------|-------|------|

**YEJA_YEJB** (3.1L)
| 466_D | 325_R | 0.90 |
|-------|-------|------|

**Figure 3.** Predicted residue–residue interactions across protein interfaces of unknown structure. Strongly co-evolving residue pairs for complexes without known structure that had at least one prediction with GREMLIN score greater than or equal to 0.85. Each row shows the residue pairs, their sequence identity and the GREMLIN score. Structure models for complexes highlighted in red are shown in *Figure 5*. Full dataset is provided with the deposited data.

proteins guided by GREMLIN predictions. Interestingly, the region that undergoes conformational change in the activating enzyme upon substrate binding (3c8f -> 3cb8 [*Becker and Kabsch, 2002*]) is in the region we predict to be in contact with PFL.

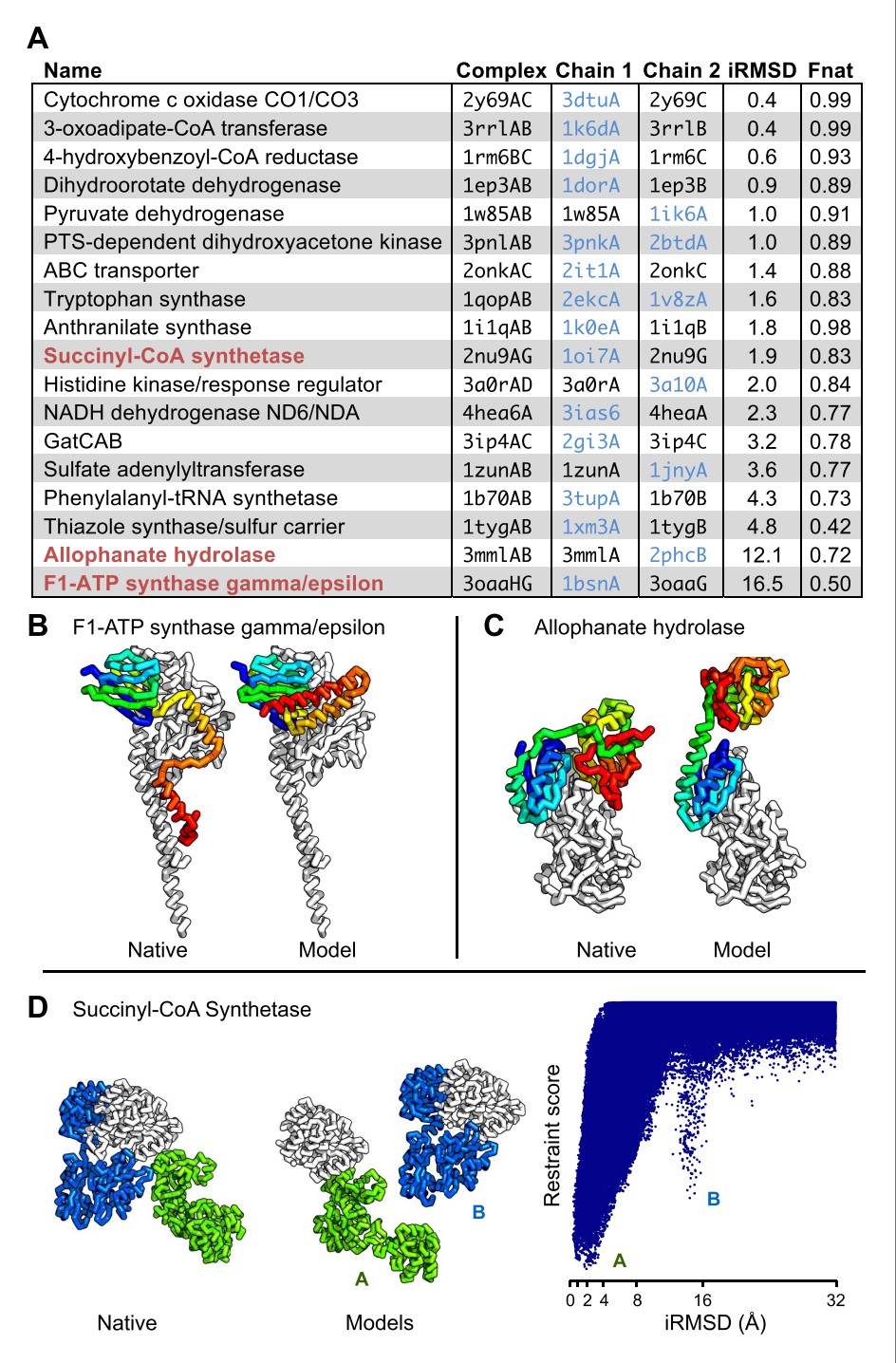

**A**

| Name | Complex | Chain 1 | Chain 2 | iRMSD | Fnat |
|---|---|---|---|---|---|
| Cytochrome c oxidase CO1/CO3 | 2y69AC | 3dtuA | 2y69C | 0.4 | 0.99 |
| 3-oxoadipate-CoA transferase | 3rrlAB | 1k6dA | 3rrlB | 0.4 | 0.99 |
| 4-hydroxybenzoyl-CoA reductase | 1rm6BC | 1dgjA | 1rm6C | 0.6 | 0.93 |
| Dihydroorotate dehydrogenase | 1ep3AB | 1dorA | 1ep3B | 0.9 | 0.89 |
| Pyruvate dehydrogenase | 1w85AB | 1w85A | 1ik6A | 1.0 | 0.91 |
| PTS-dependent dihydroxyacetone kinase | 3pnlAB | 3pnkA | 2btdA | 1.0 | 0.89 |
| ABC transporter | 2onkAC | 2it1A | 2onkC | 1.4 | 0.88 |
| Tryptophan synthase | 1qopAB | 2ekcA | 1v8zA | 1.6 | 0.83 |
| Anthranilate synthase | 1i1qAB | 1k0eA | 1i1qB | 1.8 | 0.98 |
| **Succinyl-CoA synthetase** | 2nu9AG | 1oi7A | 2nu9G | 1.9 | 0.83 |
| Histidine kinase/response regulator | 3a0rAD | 3a0rA | 3a10A | 2.0 | 0.84 |
| NADH dehydrogenase ND6/NDA | 4hea6A | 3ias6 | 4heaA | 2.3 | 0.77 |
| GatCAB | 3ip4AC | 2gi3A | 3ip4C | 3.2 | 0.78 |
| Sulfate adenylyltransferase | 1zunAB | 1zunA | 1jnyA | 3.6 | 0.77 |
| Phenylalanyl-tRNA synthetase | 1b70AB | 3tupA | 1b70B | 4.3 | 0.73 |
| Thiazole synthase/sulfur carrier | 1tygAB | 1xm3A | 1tygB | 4.8 | 0.42 |
| **Allophanate hydrolase** | 3mmlAB | 3mmlA | 2phcB | 12.1 | 0.72 |
| **F1-ATP synthase gamma/epsilon** | 3oaaHG | 1bsnA | 3oaaG | 16.5 | 0.50 |

**B** F1-ATP synthase gamma/epsilon

Native          Model

**C** Allophanate hydrolase

Native          Model

**D** Succinyl-CoA Synthetase

Native          Models

Restraint score

iRMSD (Å)

**Figure 4.** Contact guided protein–protein docking on a benchmark set of 18 protein complexes. (**A**) Structure models for each complex were generated by docking structures of its constituents, at least one of which (blue) was not from the structure of the complex guided by coevolution derived distance restraints. The interface C-alpha RMSD (iRMSD) of the structural model with the lowest energy to the experimentally determined structure and the fraction of native contacts are shown. Structure models for cases in red are shown in **B** and **C** and **D**. (**B** and **C**) Comparison between native and docked structure for the two largest failures in the benchmark: the large iRMSD is due to large conformational changes in the monomers upon docking but the interface is still modeled correctly in *Figure 4. Continued on next page*

*Figure 4. Continued*

the region not involved in conformational change. (**D**) Multiple minima in the docking landscape (right) correspond to distinct interfaces in the complex (left).

The following source data and figure supplements are available for figure 4:

**Source data 1**. Bound set.

**Figure supplement 1**. Docking landscapes showing iRMSD (x-axis) vs GREMLIN restraint score (y-axis).

**Figure supplement 2**. Bound set.

## D-methionine transport system

D-methionine transporter is an ATP-driven transport system that transports methionine. We docked the *E. coli* structure of METI (3tui, chain A and B, *Johnson et al., 2012*) with a RosettaCM model of METQ based on 3k2d (*Yu et al., 2011*). The resulting docked model is consistent with the top ranked GREMLIN predictions (*Figure 5*).

## Discussion

Our results demonstrate unequivocally that there is strong selective pressure at protein–protein interfaces beyond simple residue conservation, and that co-evolving residue pairs are nearly always in contact in the protein complex. Not all contacting residues across protein interfaces likely co-evolve nor all protein–protein interfaces. Nevertheless, as illustrated in *Figures 1 and 2*, there is clearly sufficient coevolutionary signal to significantly constrain models of a large number of protein complexes.

There is a notable contrast in the utility of intra-monomer and intersubunit predicted contacts for structure modeling. We found previously (*Kamisetty et al., 2013*) that contacts could be predicted with high accuracy for monomeric proteins, provided there were sufficient aligned sequences, but in such cases there was almost always already a structure of a family member from which comparative models could be built, limiting the utility of the predicted contacts in structure prediction (Though predicted contacts can be useful in modeling allosteric changes in protein structures [*Hopf et al., 2012*; *Morcos et al., 2013*]). In contrast, here we find that more than half of the complexes for which the protein families of the constituent subunits are sufficiently large for accurate contact prediction do not currently have three-dimensional structures. Hence, while predicted contacts can be very accurate for both monomeric globular proteins and for protein–protein complexes, they are more useful for structure modeling for the latter due to the much poorer representation of protein complexes in the PDB.

While our approach of constructing a global statistical model from paired sequence alignments is generally applicable to any taxa, the current study focuses on prokaryotes and mitochondria. Doing so allows us to largely avoid the problem of distinguishing between paralogs by exploiting the operon architecture of bacterial genomes (*Jacob et al., 2005*). Constructing paired-sequence alignments for more complex genomic architectures is more involved and requires the ability to distinguish orthologs from paralogs, the subject of active research (*Remm et al., 2001*; *Datta et al., 2009*). Protocols for generating paired sequence alignments more generally are an important area for development in this area.

## Materials and methods

### Individual alignment generation

Multiple sequence alignments were generated for each of the 4303 *E. coli* protein genes as identified by EcoGene 3.0 (*Zhou et al., 2013*) using HHblits (-n 8 -e 1E-20 -maxfilt ∞ -neffmax 20 -nodiff -realign_max ∞), and HHfilter (-id 100 -cov 75) in the HHsuite (version: 2.0.15, *Remmert et al., 2011*). To reduce redundancy, we constructed HMMs from each MSA and clustered genes based on the HHΔ (*Kamisetty et al., 2013*), a measure of HMM–HMM similarity: a pair of genes was assigned to the same cluster if the HHΔ is less than 0.5. This procedure resulted in 2340 non-redundant gene clusters.

For the benchmark set, a new alignment was generated using the sequence associated with each PDB. For the 50S ribosome and NADH dehydrogenase, we used *Thermus thermophilus* HB8 sequences

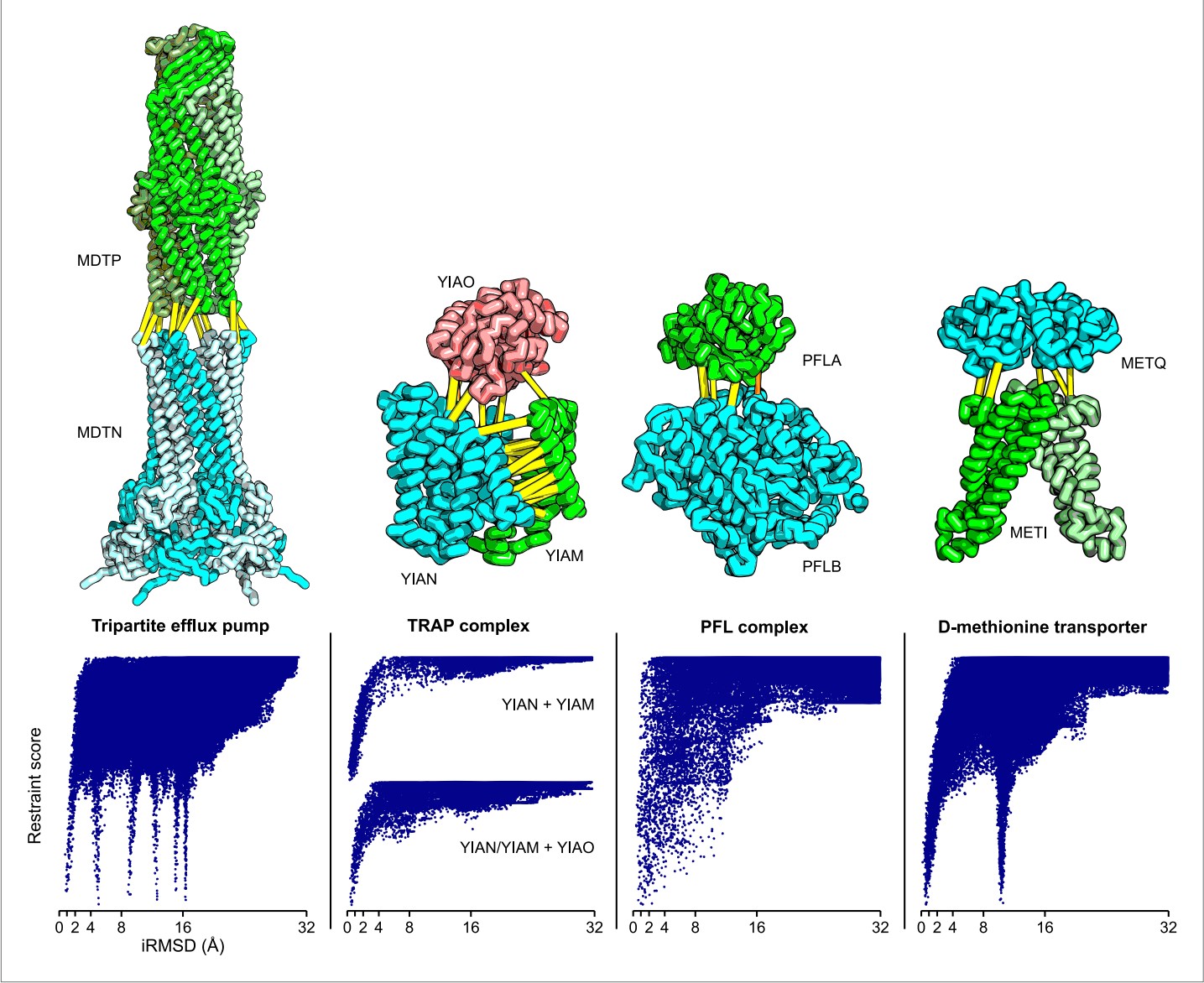

**Figure 5.** Structure models for complexes with unknown structures. Residue pairs with GREMLIN scores ≥ 0.60 are connected by yellow bars; the structures are pulled apart for clarity. For METQ-METI and PFLA-PFLB GREMLIN scores ≥ 0.3 are shown. For each docking calculation the docking energy landscape is shown, with iRMSD to the selected model on the x-axis. The multiple minima correspond to permutations of the labels on the subunits of the homo-oligomer complex. Predicted structures of each complex are provided with the deposited data.

from PDB structures 3uxr (*Bulkley et al., 2012*) and 4hea (*Baradaran et al., 2013*) respectively. For paralogous NADH dehydrogenase chains L, M, and N, we used an e-value of 1E-60 in the alignment generation protocol. In addition to complexes from the *E. coli* analysis, we also include the GatCAB amidotransferase complex in our benchmark set, using sequences from the PDB structure 3ip4 (*Nakamura et al., 2010*). For cases where the PDB sequence length was much longer than average coverage, we modified the coverage filter to 50% of query. The sequences were then realigned using clustal omega v1.2 (--iterations 2 --full-iter) (*Sievers et al., 2011*). Residues not present in the query sequence were dropped from subsequent analysis.

## Paired alignment generation

We construct alignments of paired protein sequences $[x_1, x_2, …, x_p; x_{p+1}, …, x_{p+q}]$ from the same genome with positions $1{:}p$ and $p{+}1{:}p{+}q$ corresponding to the first and second proteins respectively. We refer to such a multiple sequence alignment of paired sequences as a paired alignment.

For gene families with a single copy in each genome such as the ribosomal proteins, constructing paired alignments is straightforward as sequence pairs from the same genome can simply be concatenated. While the process of generating paired alignments in general is complicated in the presence of multiple paralogs of a gene in a single genome, in prokaryotes, co-regulated genes are often co-located on the genome into operons. We exploit this property to avoid paralogous genes when creating paired sequences by restricting to gene pairs that have small, conserved intergenic distances. A similar approach was used to construct a database of fusion proteins in prokaryotic genomes (*Suhre and Claverie, 2004*). Defining Δgene as the number of annotated genes between a gene pair, we only consider pairs with Δgene conserved in 60% of genomes and less than 20. To allow for ambiguity in annotation, if the second or third most common intergenic distance is within 1 of the mode, these gene-pairs are included in the conservation calculation. Given that most UniProt accession IDs are serially assigned in a genome (UniProt Accession), Δgene can be rapidly evaluated by looking at the difference in accession ids. The paired alignment is then filtered to reduce redundancy to 90% sequence identity and to remove positions that have more than 75% gaps.

## Identification of protein complex structures

To identify protein pairs in the same complex structure, a HMM was constructed for each *E. coli* protein using hmmbuild from the already generated HHblits alignments. We then used hmmsearch to scan PDB sequences in the S2C database (Wang et al.; Both hmmbuild and hmmsearch are part of the HMMER v3.1b package [*Eddy, 2009*]). Only hits with e-value less than 1E-10 were considered. Protein pairs found in the same complex structure (PDB file) were considered to be in contact if a Cα atom in one structure was within 12 Angstroms of a Cα atom in the other.

## Gremlin model construction from paired alignments

GREMLIN constructs a global statistical model of the paired alignment, assigning a probability to every amino-acid sequence in the paired alignment:

$$p(X_1, X_2 \ldots, X_p ; X_{p+1} \ldots X_{p+q}) = \frac{1}{Z} \exp(\Sigma_1^{p+q}[v_i(X_i) + \Sigma_{j=1}^{p+q} w_{i,j}(X_i, X_j)])$$

where, the $v_i$ are vectors encoding position-specific amino-acid propensities and the $w_{ij}$ are matrices encoding amino-acid coupling between positions i and j. These parameters are obtained from the aligned sequences by maximizing the regularized pseudo-likelihood (*Balakrishnan et al., 2011*) of the alignment as described in (*Kamisetty et al., 2013*):

$$v, w = \arg \max \Sigma_1^N \Sigma_1^{p+q} \log P(X_i \mid X_1 .. X_{i-1} X_{i+1} .. X_{p+q}) + R(v, w)$$

where, each term in the summation is a conditional distribution capturing the probability of a particular amino-acid at a position in the context of the entire protein sequence and R(v,w) is a regularization term to prevent over-fitting.

Previous approaches (*Morcos et al., 2011*; *Jones et al., 2012*) estimated v, w using an approximate moment matching approach (*Kamisetty et al., 2013*) by inverting a generalized covariance matrix. These rely on a Gaussian-like approximation to the global partition function. Unlike these approaches, estimation via the pseudo-likelihood avoids this approximation relying instead on local partition functions (*Balakrishnan et al., 2011*; *Ekeberg et al., 2013*; *Kamisetty et al., 2013*). The resulting global optimization problem can be efficiently solved using standard convex optimization techniques and provides estimates for each vector $v_i$ and matrix $w_{ij}$ (*Kamisetty et al., 2013*).

## Ranking residue pairs with gremlin scores

To reduce the $w_{ij}$ matrices to single values reflecting the strength of the coupling between positions i and j, we first compute $s_{ij}$, their vector 2-norm (the square root of the averages of the squares of the individual matrix elements). We correct for differences in $s_{ij}$ due to sequence variability at different positions using the row and column averages of these values:

$$s_{ij}^{corr} = s_{ij} - \frac{< s_{ik} >_k < s_{kj} >_k}{< s_{kl} >_{kl}}$$

where brackets indicate averages taken over the indices outside the brackets in a manner similar to that of Average Product Correction (APC, *Dunn et al., 2008*). Unlike the APC, we account for

differences in the rates of evolution in the two protein families by computing the averages only over the positions of the proteins corresponding to positions i and j: if i and j are both in the first (second) protein, the averages are computed over the positions in the first (second) protein; if i is in the first protein and j in the second, the column average is computed only over the positions of the first protein and the row average, only over the positions of the second protein. We then compute a normalized coupling strength, $ncs_{ij}$, by dividing the $s_{ij}^{corr}$ by the average of the top 3L/2 $s_{ij}^{corr}$ values across the two proteins (since there are roughly 3L/2 contacts for a protein of length L [*Kamisetty et al., 2013*; SI]).

As illustrated in *Figure 1D*, the relation between normalized coupling strength and contact frequency varies with the ratio of the number of aligned sequences to the length of the protein complex. We also observed that residues were more frequently in contact for a given coupling strength when the top score for that complex was high. To account for these dependencies, we constructed a model that estimates the probability of being in contact based on the bacterial 50S ribosomal complex:

$$\text{GremlinScore}(x, N/L) = 1/(1 + \exp(-\sigma(x - \mu)))$$

where

$$\mu = m^{N/L+1} + c$$

and x is $\sqrt{ncs_{ij}}$ for the top scoring contact in each complex and $\sqrt{ncs_{ij}}$ scaled by the Gremlin score of the top contact in all other cases. The values of m, c, and $\sigma$ (0.47, 0.96, and 9.77 respectively) were determined by a non-linear fit to the observed frequencies in the 50S ribosomal data from *Figure 1D*. This function accurately accounts for the observed contact frequencies (*Figure 1—figure supplement 1*).

## Conversion of gremlin scores to distance restraints

We converted coupling strengths into residue-pair specific distance restraints and included them in the Rosetta structure prediction program. We use sigmoidal distance restraints of the form:

$$\text{restraint}(d) = \frac{\text{weight}}{1 + \exp(-\text{slope}(d - \text{cutoff}))} + \text{intercept} \qquad (1)$$

where, d is the distance between the constrained atoms and the weight is proportional to $ncs_{ij}$. The restraints were introduced between Cβ atoms (Cα in the case of glycine) in the reduced-atom representation of Rosetta (centroid mode) and as ambiguous distance restraints (*Lange et al., 2012*) between side-chain heavy atoms (cutoff of 5.5 and slope of 4) in the full-atom stage of Rosetta. For the centroid mode, restraints used the amino acid pair specific Cβ-Cβ cutoff and slopes, as described in *Kamisetty et al., 2013* SI Table III. These distance restraints supplement the Rosetta all atom energy; the combination ensures the sampling of physically realistic structures consistent with the contact predictions.

## Comparative modeling

Comparative models were built using RosettaCM (*Song et al., 2013*) based on alignments to homologous structures generated using HHsearch (*Remmert et al., 2011*). For proteins that had missing density in regions predicted to be in contact, we used RosettaCM with co-evolution derived restraints to build the missing region before docking.

## De Novo modeling

The Rosetta ab initio protocol consists of two stages: in the initial stage ('centroid') side-chains are represented by fixed center-of-mass atoms allowing for rapid generation and evaluation of various protein-like topologies; the second stage ('full-atom') builds in explicit side-chains and carries out all atom energy minimization (*Simons et al., 1999*; *Raman et al., 2009*). YIAM, a membrane protein, was modeled with the Rosetta membrane energy function (*Yarov-Yarovoy et al., 2012*, *Barth et al., 2007*). Strong repulsive interactions (*Equation 1*, weight: −100, cutoff: 35, slope: 2 and intercept: 100) were added between the center of the extracellular regions and the center of predicted intracellular regions, and strong attractive restraints (weight:100, cutoff:35, slope:2 and intercept: 0) within predicted intracellular regions and extracellular regions, effectively constructing a membrane-like sampling space. We used the consensus output of MESSA (*Cong and Grishin, 2012*) to predict transmembrane regions. 100,000 models were generated and 20 models that best fit the restraints converged to a single cluster.

## Docking test set

Jackhammer (part of HMMER v3.1b package; *Eddy, 2009*) was used to identify a subset of 18 complexes in the benchmark set where at least one of the proteins or a close homolog had a solved structure of its *apo* form. In cases where the structure was of a homologous protein (e-value < 1E-20) and where most of the interface residues were present, we generated a structural model of the target protein using comparative modeling. We only considered cases where at least one of the structures was unbound as the bound–bound docking problem is not representative of real world docking challenges (*Betts and Sternberg, 1999*). The positive control shown in *Figure 4—source data 1* was run on all protein-pairs from the benchmark set, where at least two predicted inter contacts had a high GREMLIN score (>0.6).

## Complex assembly by protein–protein docking

For each inter restraint pair that is in the top 3/2L predictions, we used PatchDock v1.0, with clustering parameters (rmsd 0.5; discardClustersSmaller 0) (*Duhovny et al., 2002*) to generate an ensemble of conformations that were then scored using all the restraints. For tripartite efflux pump, the surface segmentation parameters were further modified (low_patch_thr 0; prune_thr 0.1; flat 1), to allow for more diverse interfaces. The top 5 models by restraint score were energy-minimized in cartesian space using both inter and intra restraints with cycles of minimization and side chain repacking using Rosetta as described in *Conway et al. (2014)*. The best scoring model by restraint score was then selected.

For fraction of native contact (Fnat) and interface root-mean-squared deviation (iRMSD) calculation, the interface residue–residue contacts are those where the minimal distance between any heavy side-chain atom is less than 5 Å. The Fnat calculation is performed as described in *Kamisetty et al. (2013)* SI Table III.

All structural figures were drawn with PyMOL (The PyMOL Molecular Graphics System, Version 1.5.0.4 Schrödinger, LLC.).

## Data Availability

The multiple sequence alignments used in the analysis and the full GREMLIN results for all the calculations described in the paper are provided at http://gremlin.bakerlab.org/complexes/ along with a web-server for paired-alignment generation, coevolution analysis and contact prediction/Rosetta restraint generation. The paired-alignments along with the PDB coordinates of the predicted structures are also available at Dryad: *Ovchinnikov et al., 2014*.

## Acknowledgements

We thank Lei Shi and David La for their comments and helpful suggestions, and Rosetta@home participants for donating their computer time.

Note added in proof

Two other studies of protein-coevolution using global statistical models have recently appeared: *Tamir et al., 2014*, and *Hopf et al., 2014*. These studies provide independent validation of the robustness of global statistical methods for prediction of protein–protein contacts.

## Additional information

### Funding

| Funder | Grant reference number | Author |
| --- | --- | --- |
| National Institutes of Health | 1R01GM092802-04 | Sergey Ovchinnikov, Hetunandan Kamisetty, David Baker |
| National Institute of Health | National Institute of General Medical Studies, P41 GM103533 | David Baker |
| Defense Threat Reduction Agency (DTRA) | N00024-10-D-6318/0024-02 | Sergey Ovchinnikov, David Baker |

The funders had no role in study design, data collection and interpretation, or the decision to submit the work for publication.

## Author contributions

SO, Conception and design, Acquisition of data, Analysis and interpretation of data, Drafting or revising the article, Contributed unpublished essential data or reagents; HK, Conception and design, Analysis and interpretation of data, Drafting or revising the article, Contributed unpublished essential data or reagents; DB, Conception and design, Analysis and interpretation of data, Drafting or revising the article

## Additional files

### Major datasets

The following dataset was generated:

| Author(s) | Year | Dataset title | Dataset ID and/or URL | Database, license, and accessibility information |
|---|---|---|---|---|
| Ovchinnikov S, Kamisetty H, Baker D | 2014 | Data from: Robust and accurate prediction of residue-residue interactions across protein interfaces using evolutionary information | http://dx.doi.org/10.5061/dryad.s00vr | Available at Dryad Digital Repository under a CC0 Public Domain Dedication. |

The following previously published datasets were used:

| Author(s) | Year | Dataset title | Dataset ID and/or URL | Database, license, and accessibility information |
|---|---|---|---|---|
| Reshetnikova L, Moor N, Lavrik O, Vassylyev DG | 2000 | PHENYLALANYL TRNA SYNTHETASE COMPLEXED WITH PHENYLALANINE | 1B70; http://dx.doi.org/10.2210/pdb1b70/pdb | Publicly available at the RCSB Protein Data Bank. |
| Wilkens S, Capaldi RA | 1998 | SOLUTION STRUCTURE OF THE EPSILON SUBUNIT OF THE F1-ATPSYNTHASE FROM ESCHERICHIA COLI AND ORIENTATION OF THE SUBUNIT RELATIVE TO THE BETA SUBUNITS OF THE COMPLEX | 1BSN; http://dx.doi.org/10.2210/pdb1bsn/pdb | Publicly available at the RCSB Protein Data Bank |
| Thoden JB, Wesenberg G, Raushel FM, Holden HM | 1999 | STRUCTURE OF CARBAMOYL PHOSPHATE SYNTHETASE COMPLEXED WITH THE ATP ANALOG AMPPNP | 1BXR; http://dx.doi.org/10.2210/pdb1bxr/pdb | Publicly available at the RCSB Protein Data Bank |
| Rebelo JM, Macieira S, Dias JM, Huber R, Romao MJ | 2000 | CRYSTAL STRUCTURE OF THE ALDEHYDE OXIDOREDUCTASE FROM DESULFOVIBRIO DESULFURICANS ATCC 27774 | 1DGJ; http://dx.doi.org/10.2210/pdb1dgj/pdb | Publicly available at the RCSB Protein Data Bank. |
| Rowland P, Larsen S | 1997 | DIHYDROOROTATE DEHYDROGENASE A FROM LACTOCOCCUS LACTIS | 1DOR; http://dx.doi.org/10.2210/pdb1dor/pdb | Publicly available at the RCSB Protein Data Bank. |
| Roberts DL, Salazar D, Fulmer JP, Frerman FE, Kim JJ-P | 1999 | ELECTRON TRANSFER FLAVOPROTEIN (ETF) FROM PARACOCCUS DENITRIFICANS | 1EFP; http://dx.doi.org/10.2210/pdb1efp/pdb | Publicly available at the RCSB Protein Data Bank. |
| Rowland P, Norager S, Jensen KF, Larsen S | 2001 | CRYSTAL STRUCTURE OF LACTOCOCCUS LACTIS DIHYDROOROTATE DEHYDROGENASE B. DATA COLLECTED UNDER CRYOGENIC CONDITIONS | 1EP3; http://dx.doi.org/10.2210/pdb1ep3/pdb | Publicly available at the RCSB Protein Data Bank. |

| Becker A, Kabsch W | 2002 | PYRUVATE FORMATE-LYASE (E.COLI) IN COMPLEX WITH PYRUVATE AND COA | 1H16; http://dx.doi.org/10.2210/pdb1h16/pdb | Publicly available at the RCSB Protein Data Bank. |
|---|---|---|---|---|
| Morollo AA, Eck MJ | 2001 | STRUCTURE OF THE COOPERATIVE ALLOSTERIC ANTHRANILATE SYNTHASE FROM SALMONELLA TYPHIMURIUM | 1I1Q; http://dx.doi.org/10.2210/pdb1i1q/pdb | Publicly available at the RCSB Protein Data Bank. |
| Kleiger G, Perry J, Eisenberg D | 2001 | 3D structure of the E1beta subunit of pyruvate dehydrogenase from the archeon Pyrobaculum aerophilum | 1IK6; http://dx.doi.org/10.2210/pdb1ik6/pdb | Publicly available at the RCSB Protein Data Bank. |
| Yamada K, Miyata T, Tsuchiya D, Oyama T, Fujiwara Y, Ohnishi T, Iwasaki H, Shinagawa H, Ariyoshi M, Mayanagi K, Morikawa K | 2002 | RuvA-RuvB complex | 1IXR; http://dx.doi.org/10.2210/pdb1ixr/pdb | Publicly available at the RCSB Protein Data Bank. |
| Vitagliano L, Masullo M, Sica F, Zagari A, Bocchini V | 2002 | Crystal structure of Sulfolobus solfataricus elongation factor 1 alpha in complex with GDP | 1JNY; http://dx.doi.org/10.2210/pdb1jny/pdb | Publicly available at the RCSB Protein Data Bank. |
| Parsons JF, Jensen PY, Pachikara AS, Howard AJ, Eisenstein E, Ladner JE | 2002 | THE CRYSTAL STRUCTURE OF AMINODEOXYCHORIS-MATE SYNTHASE FROM FORMATE GROWN CRYSTALS | 1K0E; http://dx.doi.org/10.2210/pdb1k0e/pdb | Publicly available at the RCSB Protein Data Bank. |
| Korolev S, Koroleva O, Petterson K, Collart F, Dementieva I, Joachimiak A, Midwest Center for Structural Genomics (MCSG) | 2002 | CRYSTAL STRUCTURE OF ACETATE COA-TRANSFERASE ALPHA SUBUNIT | 1K6D; http://dx.doi.org/10.2210/pdb1k6d/pdb | Publicly available at the RCSB Protein Data Bank. |
| Takahashi H, Tokunaga Y, Kuroishi C, Babayeva N, Kuramitsu S, Yokoyama S, Miyano M, Tahirov TH | 2003 | THE CRYSTAL STRUCTURE OF SUCCINYL-COA SYNTHETASE ALPHA SUBUNIT FROM THERMUS THERMOPHILUS | 1OI7; http://dx.doi.org/10.2210/pdb1oi7/pdb | Publicly available at the RCSB Protein Data Bank. |
| Weyand M, Schlichting I | 2000 | CRYSTAL STRUCTURE OF WILD-TYPE TRYPTOPHAN SYNTHASE COMPLEXED WITH INDOLE PROPANOL PHOSPHATE | 1QOP; http://dx.doi.org/10.2210/pdb1qop/pdb | Publicly available at the RCSB Protein Data Bank. |
| Unciuleac M, Warkentin E, Page CC, Dutton PL, Boll M, Ermler U | 2004 | Structure of 4-hydroxybenzoyl-CoA reductase from Thauera aromatica | 1RM6; http://dx.doi.org/10.2210/pdb1rm6/pdb | Publicly available at the RCSB Protein Data Bank. |
| Settembre EC, Dorrestein PC, Zhai H, Chatterjee A, McLafferty FW, Begley TP, Ealick SE | 2004 | Structure of the thiazole synthase/ThiS complex | 1TYG; http://dx.doi.org/10.2210/pdb1tyg/pdb | Publicly available at the RCSB Protein Data Bank. |
| Hioki Y, Ogasahara K, Lee SJ, Ma J, Ishida M, Yamagata Y, Matsuura Y, Ota M, Kuramitsu S, Yutani K, RIKEN Structural Genomics/Proteomics Initiative (RSGI) | 2005 | X-ray crystal structure of the Tryptophan Synthase b2 Subunit from Hyperthermophile, Pyrococcus furiosus | 1V8Z; http://dx.doi.org/10.2210/pdb1v8z/pdb | Publicly available at the RCSB Protein Data Bank. |

| | | | |
|---|---|---|---|
| Frank RAW, Pratap JV, Pei XY, Perham RN, Luisi BF | 2004 | THE CRYSTAL STRUCTURE OF PYRUVATE DEHYDRO-GENASE E1 BOUND TO THE PERIPHERAL SUBUNIT BINDING DOMAIN OF E2 | 1W85; http://dx.doi.org/10.2210/pdb1w85/pdb | Publicly available at the RCSB Protein Data Bank. |
| Kuzin A, Abashidze M, Vorobiev S, Forouhar F, Acton T, Ma L, Xiao R, Montelione G, Tong L, Hunt J, Northeast Structural Genomics Consortium (NESG) | 2004 | Crystal structure of Northeast Structural Genomics Target SR156 | 1XM3; http://dx.doi.org/10.2210/pdb1xm3/pdb | Publicly available at the RCSB Protein Data Bank. |
| Federici L, Du D, Walas F, Matsumura H, Fernandez-Recio J, McKeegan KS, Borges-Walmsley MI, Luisi BF, Walmsley AR | 2005 | The crystal structure of the outer membrane protein VceC from the bacterial pathogen Vibrio cholerae at 1.8 resolution | 1YC9; http://dx.doi.org/10.2210/pdb1yc9/pdb | Publicly available at the RCSB Protein Data Bank. |
| Mougous JD, Lee DH, Hubbard SC, Schelle MW, Vocadlo DJ, Berger JM, Bertozzi CR | 2006 | Crystal Structure of a GTP-Regulated ATP Sulfurylase Heterodimer from Pseudomonas syringae | 1ZUN; http://dx.doi.org/10.2210/pdb1zun/pdb | Publicly available at the RCSB Protein Data Bank. |
| Oberholzer AE, Schneider P, Bachler C, Baumann U, Erni B | 2006 | CRYSTAL STRUCTURE OF DHAL FROM E. COLI | 2BTD; http://dx.doi.org/10.2210/pdb2btd/pdb | Publicly available at the RCSB Protein Data Bank. |
| Numata T, Fukai S, Ikeuchi Y, Suzuki T, Nureki O | 2006 | crystal structure of heterohexameric TusBCD proteins, which are crucial for the tRNA modification | 2D1P; http://dx.doi.org/10.2210/pdb2d1p/pdb | Publicly available at the RCSB Protein Data Bank. |
| Asada Y, Kunishima N, RIKEN Structural Genomics/Proteomics Initiative (RSGI) | 2007 | Structural study of Project ID aq_1548 from Aquifex aeolicus VF5 | 2EKC; http://dx.doi.org/10.2210/pdb2ekc/pdb | Publicly available at the RCSB Protein Data Bank. |
| Joint Center for Structural Genomics (JCSG) | 2006 | Crystal structure of Glutamyl-tRNA(Gln) amidotransferase subunit A (tm1272) from THERMOTOGA MARITIMA at 1.80 A resolution | 2GI3; http://dx.doi.org/10.2210/pdb2gi3/pdb | Publicly available at the RCSB Protein Data Bank. |
| Lokanath NK, RIKEN Structural Genomics/Proteomics Initiative (RSGI) | 2007 | Structure of PH0203 protein from Pyrococcus horikoshii | 2IT1; http://dx.doi.org/10.2210/pdb2it1/pdb | Publicly available at the RCSB Protein Data Bank. |
| Fraser ME | 2007 | C123aT Mutant of E. coli Succinyl-CoA Synthetase Orthorhombic Crystal Form | 2NU9; http://dx.doi.org/10.2210/pdb2nu9/pdb | Publicly available at the RCSB Protein Data Bank. |
| Hollenstein K, Frei DC, Locher KP | 2007 | ABC transporter ModBC in complex with its binding protein ModA | 2ONK; http://dx.doi.org/10.2210/pdb2onk/pdb | Publicly available at the RCSB Protein Data Bank. |
| Swindell JT II, Chen L, Zhu J, Ebihara A, Shinkai A, Kuramitsu S, Yokoyama S, Fu Z-Q, Chrzas J, Rose JP, Wang B-C, Southeast Collaboratory for Structural Genomics (SECSG), RIKEN Structural Genomics/Proteomics Initiative (RSGI) | 2007 | Crystal structure of conserved uncharacterized protein PH0987 from Pyrococcus horikoshii | 2PHC; http://dx.doi.org/10.2210/pdb2phc/pdb | Publicly available at the RCSB Protein Data Bank. |
| Jormakka M, Yokoyama K, Yano T, Tamakoshi M, Akimoto S, Shimamura T, Curmi P, Iwata S | 2008 | POLYSULFIDE REDUCTASE NATIVE STRUCTURE | 2VPZ; http://dx.doi.org/10.2210/pdb2vpz/pdb | Publicly available at the RCSB Protein Data Bank. |

| | | | | |
|---|---|---|---|---|
| Ruprecht J, Yankovskaya V, Maklashina E, Iwata S, Cecchini G | 2009 | E. COLI SUCCINATE: QUINONE OXIDOREDUCTASE (SQR) WITH CARBOXIN BOUND | 2WDQ; http://dx.doi. org/10.2210/pdb2wdq/pdb | Publicly available at the RCSB Protein Data Bank. |
| Kaila VRI, Oksanen E, Goldman A, Verkhovsky MI, Sundholm D, Wikstrom M | 2011 | Bovine heart cytochrome c oxidase re-refined with molecular oxygen | 2Y69; http://dx.doi. org/10.2210/pdb2y69/pdb | Publicly available at the RCSB Protein Data Bank. |
| Yamada S, Sugimoto H, Kobayashi M, Ohno A, Nakamura H, Shiro Y | 2009 | Crystal structure of histidine kinase ThkA (TM1359) in complex with response regulator protein TrrA (TM1360) | 3A0R; http://dx.doi. org/10.2210/pdb3a0r/pdb | Publicly available at the RCSB Protein Data Bank. |
| Yamada S, Sugimoto H, Kobayashi M, Ohno A, Nakamura H, Shiro Y | 2009 | Crystal structure of response regulator protein TrrA (TM1360) from Thermotoga maritima in complex with Mg(2+)-BeF (SeMet, L89M) | 3A10; http://dx.doi. org/10.2210/pdb3a10/pdb | Publicly available at the RCSB Protein Data Bank. |
| Vey JL, Drennan CL | 2008 | 4Fe-4S-Pyruvate formate-lyase Activating Enzyme with partially disordered AdoMet | 3C8F; http://dx.doi. org/10.2210/pdb3c8f/pdb | Publicly available at the RCSB Protein Data Bank. |
| Qin L, Mills DA, Buhrow L, Hiser C, Ferguson-Miller S | 2008 | Catalytic core subunits (I and II) of cytochrome c oxidase from Rhodobacter sphaeroides complexed with deoxycholic acid | 3DTU; http://dx.doi. org/10.2210/pdb3dtu/pdb | Publicly available at the RCSB Protein Data Bank. |
| Yum S, Xu Y, Piao S, Ha N-C | 2009 | Crystal structure of E.coli MacA | 3FPP; http://dx.doi. org/10.2210/pdb3fpp/pdb | Publicly available at the RCSB Protein Data Bank. |
| Miallau L, Cascio D, Eisenberg D, TB Structural Genomics Consortium (TBSGC), Integrated Center for Structure and Function Innovation (ISFI) | 2009 | The crystal structure of the toxin-antitoxin complex RelBE2 (Rv2865-2866) from Mycobacterium tuberculosis | 3G5O; http://dx.doi. org/10.2210/pdb3g5o/pdb | Publicly available at the RCSB Protein Data Bank. |
| Sazanov LA, Berrisford JM | 2009 | Crystal structure of the hydrophilic domain of respiratory complex I from Thermus thermophilus, oxidized, 4 mol/ASU, re-refined to 3.15 angstrom resolution | 3IAS; http://dx.doi. org/10.2210/pdb3ias/pdb | Publicly available at the RCSB Protein Data Bank. |
| Nakamura A, Yao M, Tanaka I | 2009 | The high resolution structure of GatCAB | 3IP4; http://dx.doi. org/10.2210/pdb3ip4/pdb | Publicly available at the RCSB Protein Data Bank. |
| Yu S, Rhee S | 2010 | Crystal structure of Immunogenic lipoprotein A from Vibrio vulnificus | 3K2D; http://dx.doi. org/10.2210/pdb3k2d/pdb | Publicly available at the RCSB Protein Data Bank. |
| Kaufmann M, Chernishof I, Shin A, Germano D, Sawaya MR, Waldo GS, Arbing MA, Perry J, Eisenberg D, Integrated Center for Structure and Function Innovation (ISFI), TB Structural Genomics Consortium (TBSGC) | 2010 | Allophanate Hydrolase Complex from Mycobacterium smegmatis, Msmeg0435-Msmeg0436 | 3MML; http://dx.doi. org/10.2210/pdb3mml/pdb | Publicly available at the RCSB Protein Data Bank. |
| Cingolani G, Duncan TM | 2011 | Structure of the E.coli F1-ATP synthase inhibited by subunit Epsilon | 3OAA; http://dx.doi. org/10.2210/pdb3oaa/pdb | Publicly available at the RCSB Protein Data Bank. |

| | | | | |
|---|---|---|---|---|
| Shi R, McDonald L, Matte A, Cygler M, Ekiel I, Montreal-Kingston Bacterial Structural Genomics Initiative (BSGI) | 2011 | Crystal Structure of E.coli Dha kinase DhaK | 3PNK; http://dx.doi. org/10.2210/pdb3pnk/pdb | Publicly available at the RCSB Protein Data Bank. |
| Shi R, McDonald L, Matte A, Cygler M, Ekiel I, Montreal-Kingston Bacterial Structural Genomics Initiative (BSGI) | 2011 | Crystal Structure of E.coli Dha kinase DhaK-DhaL complex | 3PNL; http://dx.doi. org/10.2210/pdb3pnl/pdb | Publicly available at the RCSB Protein Data Bank. |
| Nocek B, Stein A, Marshall N, Jedrzejczak R, Babnigg G, Joachimiak A, Midwest Center for Structural Genomics (MCSG) | 2011 | Protein-protein complex of subunit 1 and 2 of Molybdopterin-converting factor from Helicobacter pylori 26695 | 3RPF; http://dx.doi. org/10.2210/pdb3rpf/pdb | Publicly available at the RCSB Protein Data Bank. |
| Nocek B, Stein A, Marshall N, Jedrzejczak R, Babnigg G, Joachimiak A, Midwest Center for Structural Genomics (MCSG) | 2011 | Complex structure of 3-oxoadipate coA-transferase subunit A and B from Helicobacter pylori 26695 | 3RRL; http://dx.doi. org/10.2210/pdb3rrl/pdb | Publicly available at the RCSB Protein Data Bank. |
| Johnson E, Nguyen PT, Rees DC | 2011 | Inward facing conformations of the MetNI methionine ABC transporter: CY5 native crystal form | 3TUI; http://dx.doi. org/10.2210/pdb3tui/pdb | Publicly available at the RCSB Protein Data Bank. |
| Safro M, Klipcan L, Moor N, Finarov I, Kessler N, Sukhanova M | 2011 | Crystal structure of human mitochondrial PheRS complexed with tRNAPhe in the active open state | 3TUP; http://dx.doi. org/10.2210/pdb3tup/pdb | Publicly available at the RCSB Protein Data Bank. |
| Bulkley D, Johnson FA, Steitz TA | 2012 | The structure of thermorubin in complex with the 70S ribosome from Thermus thermophilus. This file contains the 50S subunit of one 70S ribosome. The entire crystal structure contains two 70S ribosomes | 3UXR; http://dx.doi. org/10.2210/pdb3uxr/pdb | Publicly available at the RCSB Protein Data Bank. |
| Mancusso RL, Gregorio GG, Liu Q, Wang DN | 2012 | Crystal Structure of a bacterial dicarboxylate/sodium symporter | 4F35; http://dx.doi. org/10.2210/pdb4f35/pdb | Publicly available at the RCSB Protein Data Bank. |
| Baradaran R, Berrisford JM, Minhas GS, Sazanov LA | 2013 | Crystal structure of the entire respiratory complex I from Thermus thermophilus | 4HEA; http://dx.doi. org/10.2210/pdb4hea/pdb | Publicly available at the RCSB Protein Data Bank. |
| Broussard TC, Kobe MJ, Pakhomova S, Neau DB, Price AE, Champion TS, Waldrop GL | 2013 | Crystal Structure of Biotin Carboxyl Carrier Protein-Biotin Carboxylase Complex from E.coli | 4HR7; http://dx.doi. org/10.2210/pdb4hr7/pdb | Publicly available at the RCSB Protein Data Bank. |

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
