## [Decision Letter]

Thank you for sending your work entitled “Robust and accurate prediction of residue-residue interactions across protein interfaces using evolutionary information” for consideration at *eLife*. Despite some concerns, your article has been favorably evaluated by a Senior editor (John Kuriyan), a Reviewing editor, and 2 reviewers. To minimize delays, you should first respond by email to describe how you plan to revise the article before submitting a full revision. We are hopeful that by your submitting a detailed response this can be discussed by the editors and the reviewers, and that a revised manuscript need not then be submitted to re-review.

The Reviewing editor and the reviewers discussed their comments, and there is consensus that this is an interesting and potentially very influential manuscript using evolution methods to improve both protein structure prediction as well as protein-protein interactions. The manuscript describes the application of the pseudo-likelihood approach, first described by [1], to the determination of interacting residues across protein-protein interfaces. The coupling information is then used for the construction of docked models for a number of biologically important systems. The results seem to be very good and interesting, at least at a qualitative level. There are, however, some concerns:

1) Why were structural models built only for the unknown complexes, and why not for the ones with known structure? Quantitative results for at least some of the complexes with known structure would be helpful to better understand the accuracy of the resulting models and the impact of distance constraints.

2) There is not sufficient quantitative evidence demonstrating that the method described here is far superior to previous methods such as mfDCA and plDCA.

3) It is somewhat unfortunate that the paper as currently written overstates the novelty of the work, including unnecessary and unsubstantiated criticism of previous methods. Understandably, the authors did not want to repeat the discussion of the algorithm given previously by Balakrishnan et al. and also briefly in their PNAS paper (24). However, considering that the primary readers of *eLife* are most likely biologists, it might be useful to add a few sentences in the Introduction about the innovative nature of the pseudo-likelihood approach, and why is it better than the earlier methods. More generally, there are many places in the paper where some more explanation would help the readers. Note that *eLife* has no limitation on the length of the paper, and hence it is possible to improve readability.

---

## [Author Response]

*1) Why were structural models built only for the unknown complexes, and why not for the ones with known structure? Quantitative results for at least some of the complexes with known structure would be helpful to better understand the accuracy of the resulting models and the impact of distance constraints*.

This is an excellent suggestion. We have carried out a detailed analysis of docking results on proteins of known structure, and we will include this in the revised version. Briefly, in 30 of the 33 cases where there were at least two predicted contacts with scores greater than 0.6, the docking model with the best gremlin score was within 2 Å RMSD of the experimentally determined complex structure. We will point out that this likely overestimates the accuracy of our models of the unknown complexes, however, since docking of bound (from the structure of the complex) structures generally is more accurate than docking of unbound structures.

*2) There is not sufficient quantitative evidence demonstrating that the method described here is far superior to previous methods such as mfDCA and plDCA*.

The demonstration that the pseudo-likelihood approach is quantitatively better than previous methods was presented in the two previous papers on this approach. This is not the focus of this paper, as should be clear from the Abstract. The focus of this paper is, instead, the demonstration that co-evolution data can be used to generate reliable models of a wide range of protein protein complexes of biological interest. We go very considerably beyond previous work on this topic, which has focused on the Sensor histidine kinase+Response Regulator (SK/RR) two-component systems.

*3) It is somewhat unfortunate that the paper as currently written overstates the novelty of the work, including unnecessary and unsubstantiated criticism of previous methods. Understandably, the authors did not want to repeat the discussion of the algorithm given previously by Balakrishnan et al. and also briefly in their PNAS paper (*[24]*). However, considering that the primary readers of* eLife *are most likely biologists, it might be useful to add a few sentences in the Introduction about the innovative nature of the pseudo-likelihood approach, and why is it better than the earlier methods. More generally, there are many places in the paper where some more explanation would help the readers. Note that* eLife *has no limitation on the length of the paper, and hence it is possible to improve readability*.

The contribution of this paper is not the development of the pseudo-likelihood method, but the study of a large set of protein complexes of significant biological interest. We will provide more explanation of the innovative nature of the approach in the methods section. Where necessary, we will refer to the relevant sections of our PNAS paper (24) for technical details.